# Establishing Postnatal Growth Monitoring Curves of Preterm Infants in China: Allowing for Continuous Use from 24 Weeks of Preterm Birth to 50 Weeks

**DOI:** 10.3390/nu14112232

**Published:** 2022-05-27

**Authors:** Xin’nan Zong, Hui Li, Yaqin Zhang

**Affiliations:** Department of Growth and Development, Capital Institute of Pediatrics, Beijing 100020, China; xnzong@163.com (X.Z.); zhangyaqin81@163.com (Y.Z.)

**Keywords:** preterm infant, premature, nutrition, weight, length, head circumference, growth curve, growth reference

## Abstract

Background: Early postnatal growth monitoring and nutrition assessment for preterm infants is a public health and clinical concern. We aimed to establish a set of postnatal growth monitoring curves of preterm infants to better help clinicians make in-hospital and post-discharge nutrition plan of these vulnerable infants. Methods: We collected weight, length and head circumference data from a nationwide survey in China between 2015 and 2018. Polynomial regression and the modified LMS methods were employed to construct the smoothed weight, length and head circumference growth curves. Results: We established the P_3_, P_10_, P_25_, P_50_, P_75_, P_90_, P_97_ reference curves of weight, length and head circumference that allowed for continuous use from 24 weeks of preterm birth to 50 weeks and developed a set of user-friendly growth monitoring charts. We estimated approximate ranges of weight gain per day and length and head circumference gains per week. Conclusions: Our established growth monitoring curves, which can be used continuously without correcting gestational age from 24 weeks of preterm birth to 50 weeks, may be useful for assessment of postnatal growth trajectories, definition of intrauterine growth retardation at birth, and classification of early nutrition status for preterm infants.

## 1. Introduction

Early postnatal growth and development of preterm infants have an important impact on their future health status and disease risk. Accurate identification for early postnatal growth and nutrition deviation, such as insufficient or excessive growth, intrauterine growth retardation (IUGR) at birth, undernutrition or overnutrition, is conducive to scientific feeding guidance and nutritional management for preterm infants. Since growth curves are essential tools to monitor and evaluate early postnatal growth and nutrition deviation of preterm infants, it is of great practical value to establish scientific and user-friendly growth curves for newborn clinical practice.

In recent years, researchers have successively constructed percentile growth curves specially used for early postnatal growth monitoring for preterm infants [1,2]. Frequently-used charts, such as the Fenton 2013 or the Olsen 2015 preterm growth charts, did not include a Chinese cohort in their development [3]. Considering there are racial/ethnic differences in children’s growth and development worldwide [4,5], foreign growth curves may not fully reflect early postnatal growth and nutrition status of Chinese preterm infants [6,7]. In fact, China has also lacked a set of user-friendly growth curves for early postnatal growth monitoring and nutrition assessment for preterm infants. Therefore, it is necessary to develop postnatal growth monitoring curves of preterm infants in China. 

We aimed to establish a set of postnatal growth monitoring curves of preterm infants based on a nationwide growth and development survey in China between 2015 and 2018 to better help clinicians make in-hospital and post-discharge nutrition plan of these vulnerable infants.

## 2. Materials and Methods

### 2.1. Study Design and Data Source

We established postnatal growth monitoring curves of preterm infants allowing for continuous use from 24 weeks of preterm birth to 50 weeks based on two sets of growth reference values. First, the Newborn Growth Curves contribute to percentile reference values of weight, length and head circumference by sex and gestational age from 24 weeks to 42 weeks. Second, the Child Growth Curves contribute to percentile reference values of weight, length and head circumference by sex and age from 37 weeks of full-term birth to 50 weeks (equivalent to postnatal 10 weeks of full-term birth). Establishing postnatal growth monitoring curves based on existing percentile values of the growth curves was confirmed to be scientific and feasible by a series of studies by Fenton and colleagues [1,8].

The Newborn Growth Curves were constructed based on a large-scale cross-sectional sample of 24,375 newborn babies with a gestational age of 24–42 weeks in thirteen cities in China from June 2015 to November 2018 [9]. The Child Growth Curves were constructed based on a large-scale cross-sectional sample of 83,628 children from full-term birth to seven years old in nine cities in China from June to November 2015 [10]. Both the sample population of the Newborn Growth Curves and the sample population of the Child Growth Curves came from the fifth National Survey on the Physical Growth and Development of Children in China (NSPGDC), which was conducted under the framework of the nine cities (i.e., Beijing, Harbin, Xi’an, Shanghai, Nanjing, Wuhan, Guangzhou, Fuzhou and Kunming). In essence, these two samples belonged to a single reference population with high homogeneity. Considering that the actual number of early preterm babies was very small, the data collection time was appropriately extended and the four survey sites were added in Tianjin, Shenyang, Changsha and Shenzhen surrounding the nine cities of the NSPGDC when collecting preterm babies with <32 weeks of gestation [11]. In the NSPGDC, the inclusion and exclusion criteria of the study subjects can be seen in references [12,13].

### 2.2. Process of Establishing Postnatal Growth Monitoring Curves

#### 2.2.1. Selecting Seven Main Percentile Curves

To ensure goodness of fit of the growth curves, seven frequently-used main percentile curves (P_3_, P_10_, P_25_, P_50_, P_75_, P_90_, P_97_) were selected as the initial curves to be smoothed according to statistical accuracy and international practice [1,14].

#### 2.2.2. Selecting Age Target Points of Initial Percentile Curves

Since establishing postnatal growth monitoring curves of preterm infants mainly involved reasonable merge between the full-term parts of the Newborn Growth Curves and the Child Growth Curves, we referred to the real practice of selecting age target points in establishing the Fenton 2013 growth monitoring curves [1] and observed the specific curve shapes of the Newborn Growth Curves and the Child Growth Curves. Several age target points of initial percentile curves were selected in this study, including 36, 38, 40, 42, 44 and 46 weeks.

#### 2.2.3. Obtaining Preliminarily Smoothed Percentile Values

Based on the curve shapes of the Newborn Growth Curves and the Child Growth Curves, the initial values of each of seven main percentiles of weight, length and head circumference were obtained at the age target points between 36 weeks and 46 weeks by an observation method to make the merged curves look smooth, and the percentile values at 24–35 weeks and at 47–50 weeks remained unchanged. Polynomial regression was employed to fit the initial values of each of seven main percentiles of weight, length and head circumference by sex, then a series of smoothed predicted percentile values were reobtained between 36 weeks and 46 weeks. Finally, we obtained preliminarily smoothed seven main percentile values of weight, length and head circumference for boys and girls at each week between 24 weeks and 50 weeks.

#### 2.2.4. Obtaining Smoothed L, M and S Parameters

L, M and S parameters of weight, length and head circumference by sex and age were fitted based on the above preliminarily smoothed seven main percentile values using the nonlinear equation from the modified LMS methods [15,16].

#### 2.2.5. Obtaining Standardized Smoothed Percentile Values

Standardized smoothed percentile values of weight, length and head circumference by sex and age were calculated based on the above smoothed L, M and S parameters using the following nonlinear equation from the modified LMS methods:C_100α_(t) = M(t)[1 + L(t)S(t)z_α_]^1/L(t)^(1)
where C_100__α_(t) is the centile curve plotted against age t, z_α_ is the normal equivalent deviate for the centile (for example when α = 0.97 corresponding to P_97_, z_α_ = 1.88), and L(t), M(t) and S(t) are the fitted smoothed curves plotted against age.

#### 2.2.6. Assessing the Fitted Performance of the Standardized Smoothed Percentile Curves

Standardized smoothed seven main percentile curves of weight, length and head circumference were compared with the preliminarily smoothed seven main percentile curves between 24 weeks and 50 weeks. A small difference (i.e., weight < 0.01 kg, length < 0.1 cm, head circumference < 0.1 cm) by sex and age was regarded as a good fit.

### 2.3. Fenton 2013 Growth Monitoring Curves and INTERGROWTH Growth Curves

To better understand and apply postnatal growth curves of preterm infants, we compared the established postnatal growth monitoring curves with the Fenton 2013 growth monitoring curves [1] and the INTERGROWTH growth curves [2].

The Fenton 2013 curves were generated based on newborn percentile values from European and American developed countries and children’s percentile values from the WHO Child Growth Standards, including weight, length and head circumference growth curves for boy and girls [1]. Specifically, the weight curves came from newborn weight percentile values of the 22–40 weeks of gestation from six countries (i.e., Germany (1995–2000), United States (1998–2006), Italy (2005–2007), Australia (1991–1994), Scotland (1998–2003) and Canada (1994–1996)) and children’s weight percentile values of full-term 40 weeks to postnatal 10 weeks from the WHO Child Growth Standards. The length and head circumference curves came from newborn length and head circumference percentile values of the 23–40 weeks of gestation from two countries (i.e., United States (1998–2006) and Italy (2005–2007)) and children’s length and head circumference percentile values of full-term 40 weeks to postnatal 10 weeks from the WHO Child Growth Standards. The Fenton 2013 curves covered the age range of 24 weeks of preterm birth to 50 weeks.

The INTERGROWTH curves were generated based on a population-based longitudinal data of 201 preterm babies with 26–36 weeks of gestation from eight locations worldwide: Pelotas, Brazil; Turin, Italy; Muscat, Oman; Oxford, UK; Seattle, WA, USA; Shunyi County, Beijing, China; central Nagpur, India; and Parklands suburb, Nairobi, Kenya between 2009 and 2014 [2]. Weight, length and head circumference were measured within 12 h of birth and thereafter every 2 weeks in the first 2 months and every 4 weeks until postnatal age 8 months; a total of 1759 sets of measures were recorded. The INTERGROWTH curves included weight, length and head circumference growth curves for boys and girls with age range of 27 weeks of preterm birth to 64 weeks.

### 2.4. Statistical Analysis

Polynomial regression was employed to fit the initial values of each of seven main percentiles of weight, length and head circumference for boys and girls between 36 weeks and 46 weeks to obtain preliminarily smoothed P_3_, P_10_, P_25_, P_50_, P_75_, P_90_, P_97_ values. A nonlinear equation of the modified LMS methods was used to fit seven preliminarily smoothed main percentile values of weight, length and head circumference for boys and girls between 24 weeks and 50 weeks to obtain the smoothed L, M and S parameters. Standardized smoothed percentile and Z-score values of weight, length and head circumference by sex and age were finally generated based on the smoothed L, M and S parameters. R-square was used to assess goodness of fit of the growth curves. The P_3_, P_10_, P_50_, P_90_, P_97_ values of the established postnatal growth monitoring curves were compared with the corresponding percentile curves of the Fenton 2013 curves [1] and the INTERGROWTH curves [2]. Statistical analysis was performed by SAS 9.4 (SAS Institute Inc., Cary, NC, USA).

## 3. Results

### 3.1. Established Postnatal Growth Monitoring Curves and their Comparison with Newborn Growth Curves and the Child Growth Curves

Based on seven main percentile values of weight, length and head circumference by sex and week from the Newborn Growth Curves (extract data from 24 weeks of gestation to 42 weeks) and the Child Growth Curves (extract data from 37 weeks of full-term birth to 50 weeks), after a smoothing process of polynomial regression and the modified LMS methods, we fitted postnatal growth monitoring curves of weight, length and head circumference for boys and girls that covered the age range from 24 weeks of preterm birth to 50 weeks. Overall, the established postnatal growth monitoring curves can be fairly well linked to the Newborn Growth Curves and the Child Growth Curves (Figure 1A–C). The established postnatal growth curves were not very different from the Newborn Growth Curves at 24–36 weeks and the Child Growth Curves at 46–50 weeks, but higher than both the Newborn Growth Curves and the Child Growth Curves at 37–45 weeks.

### 3.2. Postnatal Growth Monitoring Curve Allowing for Continuous Use from 24 Weeks of Preterm Birth to 50 Weeks

We obtained standardized smoothed P_3_, P_10_, P_25_, P_50_, P_75_, P_90_, P_97_ reference values of weight, length and head circumference for preterm boys and girls that allowed for continuous use from 24 weeks of preterm birth to 50 weeks. Data including percentile and Z-score reference values are available for research upon request.

Based on the established postnatal growth monitoring curves, the approximate ranges of the “growth velocity” were estimated as follow: weight gain of (17–18) ± 2 g/kg/d at 24–36 weeks, (10−11) ± 2 g/kg/d at 37–42 weeks, (6−7) ± 1 g/kg/d at 43–50 weeks; length gain of (1.3–1.5) cm/w at 24–32 weeks, (1.1−1.3) cm/w at 33–36 weeks, (0.8−0.9) cm/w at 37–50 weeks; head circumference of (0.9−1.0) cm/w at 24–32 weeks, (0.6−0.7) cm/w at 33–36 weeks, 0.5 cm/w at 37–50 weeks.

In order to facilitate actual clinical application, we drew a set of user-friendly postnatal growth monitoring charts for boys and girls which were composed of the frequently-used seven main percentile curves of weight, length and head circumference in the charts (Figure 2 and Figure 3).

### 3.3. Comparison of the Established Postnatal Growth Monitoring Curves with the Fenton 2013 Growth Curves

Overall, the curve shapes and trajectories of the established postnatal weight, length and head circumference growth monitoring curves were consistent with those of the Fenton 2013 curves (Figure 4A–C). The P_3_ and P_10_ curves of weight were somewhat higher than the corresponding percentile curves of the Fenton 2013 curves, which seemed to be obvious at 27–34 weeks and at 40–50 weeks. The P_3_ and P_10_ curves of length were slightly higher than the corresponding percentile curves of the Fenton 2013 curves at 34–42 weeks. The growth curves of head circumference were slightly lower than the corresponding percentile curves of the Fenton 2013 curves at 37–45 weeks.

### 3.4. Comparison of Established Postnatal Growth Monitoring Curves with the INTERGROWTH Growth Curves

Overall, the curve shapes and trajectories of the established postnatal weight, length and head circumference growth monitoring curves were similar to those of the INTERGROWTH curves between 27 and 50 weeks (Figure 5A–C). The P_3_ and P_10_ curves of weight were higher, the P_97_ and P_90_ curves of length were higher, and the P_3_ and P_10_ curves of head circumference were lower than the corresponding percentile curves of the INTERGROWTH curves.

## 4. Discussion

Regarding early postnatal growth monitoring for preterm infants, traditionally, newborn growth curves are first used, and then child growth curves are used when preterm infants grow to full-term age through a correction of gestational age. However, this practice leads to the depiction of growth monitoring trajectories of preterm infants on different growth curves, which is not conducive to quickly and accurately identifying early growth and nutrition deviation for pediatricians and child health physicians. In this study, we established a new set of postnatal growth monitoring curves for preterm infants by deeply integrating newborn growth curves and child growth curves. Using the established postnatal growth curves is essentially the same as the traditional evaluation process, but simplifies the evaluation process, as it enables early postnatal growth monitoring trajectories of preterm infants to continuously plot on a single chart without correcting gestational age from 24 weeks of preterm birth to 50 weeks. With the help of this simple and convenient practical tool, clinicians can timely and accurately find early growth and nutrition deviation of preterm infants (such as insufficient or excessive growth, IUGR at birth, and undernutrition or overnutrition), so as to better guide early scientific feeding and nutrition management for these vulnerable infants.

Our established curves were consistent with the Fenton 2013 curves in the smoothing methods and process of developing the growth curves. The curve shapes and trajectories of our established curves were similar to those of the Fenton 2013 curves that were generated based on a systematic review and meta-analysis. The main differences of these two curves were reflected in the low percentile curves (such as P_3_, P_10_) of weight at 27–34 weeks and at 40–50 weeks. The main reason was that the Fenton 2013 curves were linked to the WHO Child Growth Standards, including 2.3% of “unhealthy” full-term infants with low birth weight (<2.5 kg) [17], which may cause an overall slightly lighter for the Fenton 2013 weight curves than our established weight curves. A recent study in China showed that postnatal weight gain of late preterm infants was higher than the Fenton 2013 curves [6], suggesting that our established curves may be more appropriate in monitoring and evaluating early postnatal growth and nutrition deviation of Chinese preterm infants. Future research needs to further directly evaluate the applicability of our established curves in Chinese population.

The curve shapes and trajectories of our established curves were similar to those of the INTERGROWTH curves that were constructed based on longitudinal data of preterm infants [2]. The main differences of these two curves were reflected in weight, length and head circumference curves at 27–30 weeks, and low percentile curves (such as P_3_, P_10_) of weight at 37–50 weeks. The main reasons may be attributed to the difference of study design and data type (cross-sectional data vs. longitudinal data) and very small sample sizes of the INTERGROWTH curves at 27–32 weeks. Some recent studies showed that the Fenton 2013 curves and the INTERGROWTH curves had various differences depending on the evaluated populations in classification of extrauterine growth restriction [18,19,20], which indirectly suggests that it may be more appropriate to assess early postnatal growth and nutrition deviation of preterm infants in a population using the growth curves generated from the same racial/ethnic and living environmental population. A recent study on comparison of the growth trajectories of Chinese preterm infants with the INTERGROWTH curves pointed out that although the international growth curves were recommended to be applied globally, there are still limitations in practical application due to the differences in racial/ethnic groups, economic developmental level and social culture across the world, and thus the nationally representative growth curves of preterm infants in China should be established as soon as possible [7].

Postnatal growth of preterm infants reflects the growth and development status of newborns in the extrauterine environment after leaving the mothers. However, the growth level of newborns at birth reflects the growth and development status of the fetus in utero, and the newborn growth curves based on newborn data at birth show a rapid increase in preterm period and a slowing increase in full-term period. With the acceleration of catch-up growth of preterm infants after birth, the postnatal growth level of preterm infants at 37–42 weeks may be slightly higher than that of full-term infants at birth [2,21,22,23]. As expected, our established curves were higher than the Newborn Growth Curves at the full-term period. In practical applications, our established curves may be useful for assessment of postnatal growth trajectories, definition of IUGR at birth, and classification of early nutrition status for preterm infants, but not appropriate for classification of small for gestational age or large for gestational age at the birth of full-term infants.

Our study has several strengths. First, the study was based on a large-scale nationally representative sample of newborns and children in modern China. Second, our established curves were the first set of postnatal growth monitoring curves of preterm infants in China that allowed for continuous use from early preterm birth to 3 months or so after birth. Third, we estimated a relatively reasonable approximate range of weight gain per day and length and head circumference gains per week from 24 weeks of preterm birth to 50 weeks. However, our study has also a limitation. Our established curves were generated based on cross-sectional data and advanced statistical methods, which realized a good connection between newborn growth curves and child growth curves. From the curve shapes and trajectories of our established curves, they seem realistic, but need to be further verified using follow-up data of preterm infants and in real clinical practice.

## 5. Conclusions

Our established postnatal growth monitoring curves are applicable to preterm infants and do not require correction for gestational age in actual use. Early postnatal growth trajectories of preterm infants can be continuously depicted to 50 weeks on this set of the growth charts, and some common growth and nutrition deviations of preterm infants, such as insufficient or excessive growth, IUGR at birth, undernutrition or overnutrition, can be screened out in time with the help of this convenient, clinical, practical tool. In addition, our study has added more evidence to better help clinicians make in-hospital and post-discharge nutrition plans for vulnerable infants, especially for preterm infants with similar ethnic and genetic backgrounds in the Asian part of the world. Further studies should be conducted to evaluate the long-term outcomes (such as neurologic and cardio-metabolic morbidities) of vulnerable infants with IUGR at birth or early overweight/obesity identified by our established postnatal growth monitoring curves.

## Figures and Tables

**Figure 1 nutrients-14-02232-f001:**
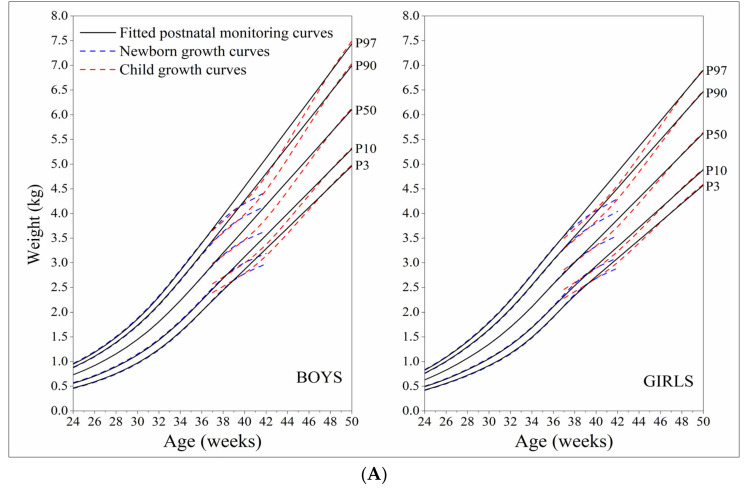
Comparison of the 3rd, 10th, 50th, 90th, 97th percentile curves among fitted postnatal growth monitoring curves, newborn growth curves, and child growth curves; (**A**) weight; (**B**) length; (**C**) head circumference.

**Figure 2 nutrients-14-02232-f002:**
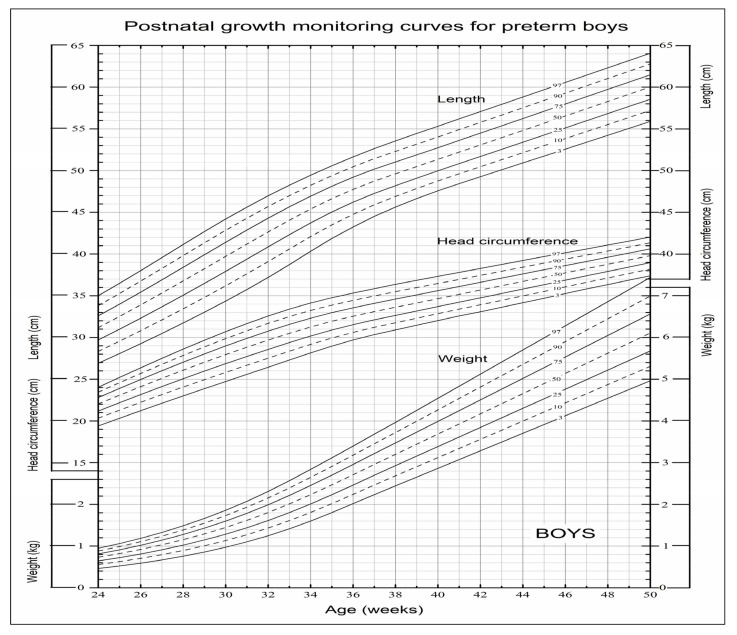
Postnatal growth monitoring charts for preterm boys in China.

**Figure 3 nutrients-14-02232-f003:**
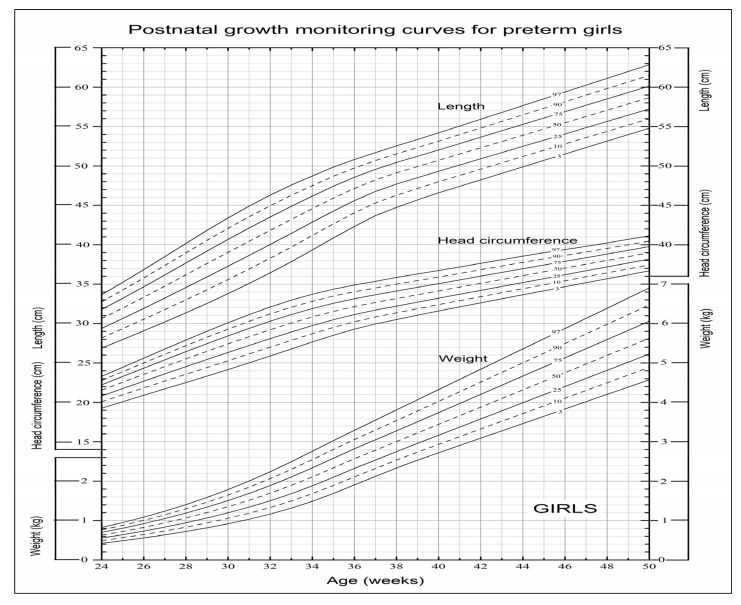
Postnatal growth monitoring charts for preterm girls in China.

**Figure 4 nutrients-14-02232-f004:**
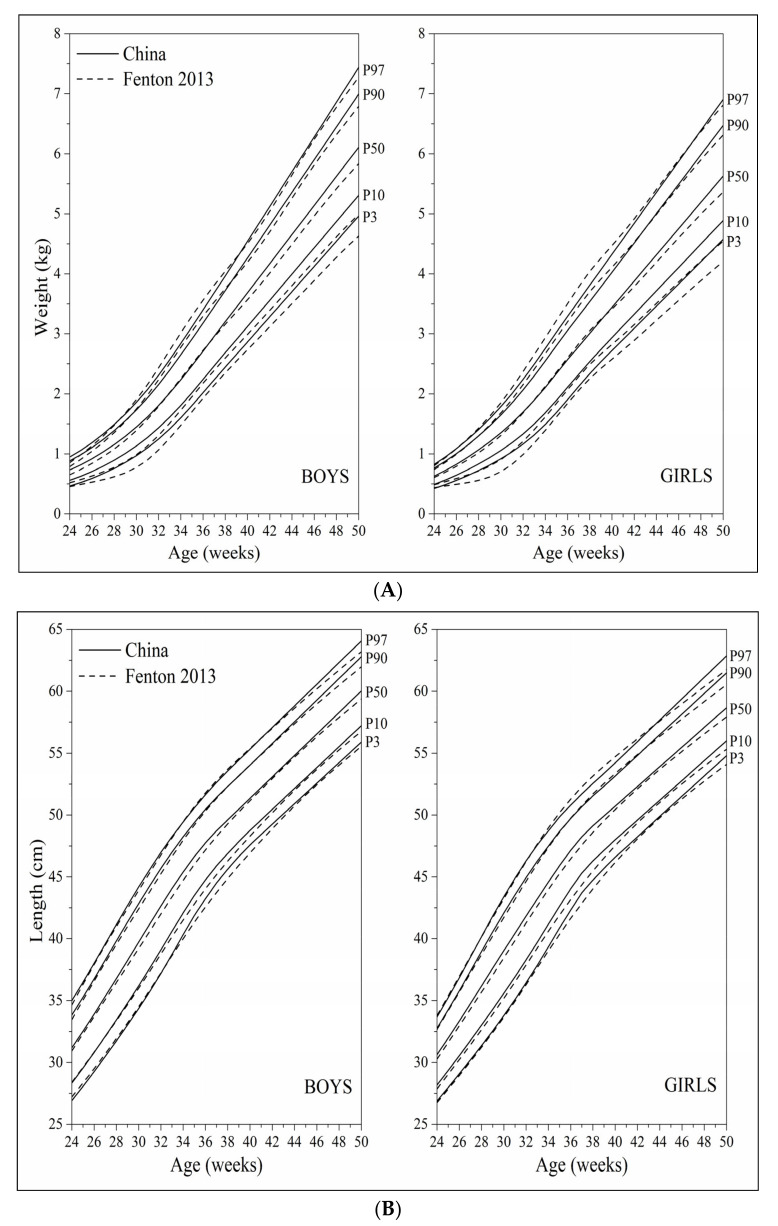
Comparison of the 3rd, 10th, 50th, 90th, 97th percentile curves between the China curves and the Fenton 2013 curves; (**A**) weight; (**B**) length; (**C**) head circumference.

**Figure 5 nutrients-14-02232-f005:**
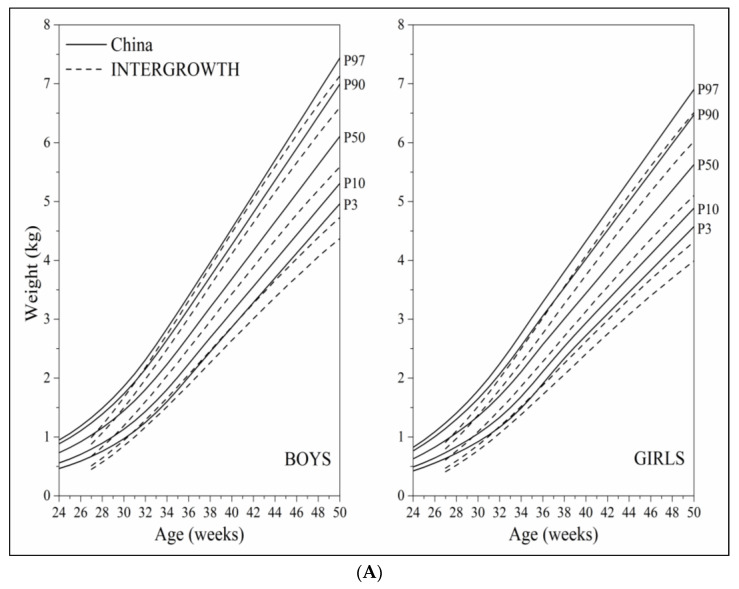
Comparison of the 3rd, 10th, 50th, 90th, 97th percentile curves between the China curves and the INTERGROWTH curves; (**A**) weight; (**B**) length; (**C**) head circumference.

## Data Availability

The data supporting the conclusions of this article will be made available from the corresponding author upon request.

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
