# Peer review of "Establishing Postnatal Growth Monitoring Curves of Preterm Infants in China: Allowing for Continuous Use from 24 Weeks of Preterm Birth to 50 Weeks"

_nutrients, 2022, doi:10.3390/nu14112232_

Round 1

Reviewer 1 Report

Thank you for allowing me to review this manuscript.  Assessment of most optimal growth in preterm infants is always applicable to current clinical practice.  My assessment of the manuscript and suggestions for enhancement include:

  • Line 3 under “Introduction”: recommend changing “excessive catch-up growth” to “excessive growth”.
  • I suggest not using the term EUGR ("Extrauterine growth restriction" and "postnatal growth failure" are misnomers for preterm infants by Fenton et al., 2020)
  • Many countries use the 2013 Fenton or the 2015 Olsen preterm growth charts. I suggest you include in your introduction that these did not include a Chinese cohort in their development and provide appropriate references.
  • Can you include how many infant’s data was used for the development of this growth chart?
  • How did you include/exclude birth growth data for infants impacted growth-wise by conditions such as preeclampsia, IUGR, uncontrolled diabetes, etc?
  • Can you comment on if you think the Chinese growth curves are appropriate to plot slightly higher or lower than other growth charts, such as the Fenton?
  • Can you please alter under the limitations section, how your growth curves allow for growth monitoring from “early preterm birth to the first 3 months after birth.” It depends on the gestational age at birth for how long infants can be monitored on this (so more or less than 3 months after birth).
  • Do your growth curves have associated z-scores?
  • Overall, I think the development of this growth chart (to mimic the Fenton growth chart) is novel and applicable for use in a Chinese preterm population. I agree that future analysis will compare is applicability in clinical practice.  Likewise, if this becomes a widely used growth chart in China, I suggest creating a shortened name (similar to something like the Olsen or Fenton) to refer to it as.

Author Response

Response to reviewer 1

Thank you for allowing me to review this manuscript.  Assessment of most optimal growth in preterm infants is always applicable to current clinical practice.  My assessment of the manuscript and suggestions for enhancement include:

Line 3 under “Introduction”: recommend changing “excessive catch-up growth” to “excessive growth”.

Response #: Revised to“excessive growth”through the whole draft according to your recommendation.

I suggest not using the term EUGR ("Extrauterine growth restriction" and "postnatal growth failure" are misnomers for preterm infants by Fenton et al., 2020)

Response #: Accepted your suggestion and removed relevant description.

Many countries use the 2013 Fenton or the 2015 Olsen preterm growth charts. I suggest you include in your introduction that these did not include a Chinese cohort in their development and provide appropriate references.

Response #: Added this content in revised draft.(i.e.,The frequently-used charts, such as the Fenton 2013 or the Olsen 2015 preterm growth charts, did not include a Chinese cohort in their development (Clark RH,et al, Clin Perinatol, 2014.)

Can you include how many infant’s data was used for the development of this growth chart?

Response #: Added relevant data /sample size in “Study Design and Data Source”section.( 24,375 newborn babies with a gestational age of 24-42 weeks and 83,628 children from full-term birth to seven years old)

How did you include/exclude birth growth data for infants impacted growth-wise by conditions such as preeclampsia, IUGR, uncontrolled diabetes, etc?

Response #: Added inclusion and exclusion description seen in the references (Zong XN, et al, Sci Rep, 2021; Zhang YQ, Am J Phys Anthropol, 2017).

For example, detailed inclusion and exclusion for birth growth data below: Single naturally conceived live births with a GA of 24-42 weeks were included. Infants who were not healthy or whose mothers were at high health risk were excluded according to the following exclusion criteria: â‘  unclear GA; â‘¡ severe congenital malformation at birth or known chromosomal abnormality; â‘¢ edema or hematoma during physical measurement; â‘£ parents of non-Chinese origin; ⑤ mothers were not permanent residents in surveyed cities and lived in surveyed cities for <2 years; â‘¥ maternal height <145 cm; ⑦maternal age <18 years or >40 years; ⑧ mothers who were smoking, alcoholic or drug dependent over the three months before or during pregnancy; ⑨ mothers who had continuously taken adrenal cortex hormones or other immunosuppressive agents for >1 month during pregnancy; â‘© mothers of full-term babies with any of the following conditions during pregnancy: severe anemia (Hb≦60g/L), gestational diabetes, preeclampsia, eclampsia, hyperthyroidism or hypothyroidism, heart and kidneys dysfunction,  chronic hypertension; ⑪ mothers of preterm babies with any of the following conditions during pregnancy: severe anemia (Hb≦60g/L), gestational diabetes that cannot be effectively controlled by diet and exercise intervention, severe preeclampsia, eclampsia, hyperthyroidism or hypothyroidism that cannot be effectively controlled by drug therapy, severe heart and kidneys dysfunction.

Can you comment on if you think the Chinese growth curves are appropriate to plot slightly higher or lower than other growth charts, such as the Fenton?

Response #: We discuss this difference in the 2nd paragraph of Discussion section.( i.e., The main differences of these two curves were reflected in the low percentile curves (such as P3, P10) of weight at 27-34 weeks and at 40-50 weeks. The main reason was that the Fenton 2013 curves were linked to the WHO Child Growth Standards including 2.3% of “unhealthy” full-term infants with low birth weight (<2.5 kg) that may cause an overall slightly lighter for the Fenton 2013 weight curves than our established weight curves. A recent study in China showed that postnatal weight gain of late preterm infants was higher than the Fenton 2013 curves, suggesting that our established curves may be more appropriate to monitor and evaluate early postnatal growth and nutrition deviation of Chinese preterm infants. Future researches need to further directly evaluate the applicability of our established curves in Chinese population.)

Can you please alter under the limitations section, how your growth curves allow for growth monitoring from“early preterm birth to the first 3 months after birth.” It depends on the gestational age at birth for how long infants can be monitored on this (so more or less than 3 months after birth).

Response #: Revised to“from early preterm birth to 3 months or so after birth.”

Do your growth curves have associated z-scores?

Response #: Our established curves included both percentile and z-score curves. Individual z-scores can be calculated based on the L, M and S parameters.

Overall, I think the development of this growth chart (to mimic the Fenton growth chart) is novel and applicable for use in a Chinese preterm population. I agree that future analysis will compare is applicability in clinical practice.  Likewise, if this becomes a widely used growth chart in China, I suggest creating a shortened name (similar to something like the Olsen or Fenton) to refer to it as.

Response #: Thank you for your valuable comments and suggestion. According to the international practice, it maybe be shortened to the 2022 Zong (or the 2022 Zong preterm growth charts).

Reviewer 2 Report

This article explored the postnatal growth curves of preterm Chinese infants. The study is well-designed, and the results are very clear and presented well. I don't see major concerns. 

Author Response

Response to reviewer 2

Comments and Suggestions for Authors

This article explored the postnatal growth curves of preterm Chinese infants. The study is well-designed, and the results are very clear and presented well. I don't see major concerns. 

Response #: Thank you for your valuable comments.

This manuscript is a resubmission of an earlier submission. The following is a list of the peer review reports and author responses from that submission.